# Is There an Ideal Diet to Protect against Iodine Deficiency?

**DOI:** 10.3390/nu13020513

**Published:** 2021-02-04

**Authors:** Iwona Krela-Kaźmierczak, Agata Czarnywojtek, Kinga Skoracka, Anna Maria Rychter, Alicja Ewa Ratajczak, Aleksandra Szymczak-Tomczak, Marek Ruchała, Agnieszka Dobrowolska

**Affiliations:** 1Department of Gastroenterology, Dietetics and Internal Diseases, Poznan University of Medical Sciences, Heliodor Swiecicki Hospital, 60-355 Poznan, Poland; krela@op.pl (I.K.-K.); a.m.rychter@gmail.com (A.M.R.); alicjaewaratajczak@gmail.com (A.E.R.); aleksandra.szymczak@o2.pl (A.S.-T.); agdob@ump.edu.pl (A.D.); 2Department of Endocrinology, Metabolism and Internal Medicine, Poznan University of Medical Sciences, 60-355 Poznan, Poland; agata.rat@wp.pl (A.C.); mruchala@ump.edu.pl (M.R.); 3Department of Pharmacology, Poznan University of Medical Sciences, 60-806 Poznan, Poland

**Keywords:** iodine deficiency, diet, iodine, vegetarian diet, alternative diet

## Abstract

Iodine deficiency is a global issue and affects around 2 billion people worldwide, with pregnant women as a high-risk group. Iodine-deficiency prevention began in the 20th century and started with global salt iodination programmes, which aimed to improve the iodine intake status globally. Although it resulted in the effective eradication of the endemic goitre, it seems that salt iodination did not resolve all the issues. Currently, it is recommended to limit the consumption of salt, which is the main source of iodine, as a preventive measure of non-communicable diseases, such as hypertension or cancer the prevalence of which is increasing. In spite of the fact that there are other sources of iodine, such as fish, seafood, dairy products, water, and vegetables, the high consumption of processed food with a high content of unionised salt, alternative diets or limited salt intake can still lead to iodine deficiency. Thus, iodine deficiency remains a relevant issue, with new, preventive solutions necessary. However, it appears that there is no diet which would fully cover the iodine requirements, and iodine food supplementation is still required.

## 1. Introduction

Iodine is absorbed in the stomach and the small intestine where it is transported via sodium/iodide symporter (NIS) and pendrin to the thyroid gland, and then stored in the follicular cells. Subsequently, in the presence of hydrogen peroxide (H_2_O_2_), iodine ions (I^−^) are oxidised by thyroid peroxidase. Then, the tyrosine residues are iodinated—first, at position 3, which forms monoiodotyrosine (MIT), and then at position 5 to form diiodotyrosine (DIT). MIT and DIT in a coupling reaction form thyroxine (T_4_) and triiodothyronine (T_3_) hormones [1,2]. As a trace element, iodine is essential in human nutrition, mostly due to its role in the thyroid hormone synthesis. Moreover, both T_4_ and T_3_ are involved in the regulation of metabolic processes in the human body and are responsible for the optimal growth of the central nervous system and brain [3,4]. Iodine also serves as an antioxidant and exhibits the protective effects of inflammatory states and cancer [5]. Iodine deficiency constitutes a global issue, and is reflected by urinary iodine concentration (UIC) < 100 µg/day [3]. It is prevalent in the eastern Mediterranean, Asia, Eastern Europe, and Africa. Additionally, mild iodine deficiency is observed in Australia, Great Britain, and New Zealand, as well as in specific groups, such as vegans or vegetarians [6]. In fact, proper iodine intake is crucial among pregnant women, as iodine deficiency is the leading cause of mental retardation in children [7]. Since 1993, World Health Organization (WHO) and United Nations Children’s Fund (UNICEF) recommend a universal salt iodisation. Currently, around 70% of households in over 120 countries have access to iodised salt—in 1990, only less than 10% had this advantage [8]. Iodine is absorbed in 90% by stomach and duodenum and is mostly delivered from the fortified salt, and other sources such as fish, seafood, dairy, water, eggs, broccoli, peas, or spinach [4,9]. However, it is vital to notice that different concentrations of iodine in food products have been noticed, which is presumably associated with different growth environments [10]. Both the deficiency and excess of iodine can lead to an impaired thyroid gland function and, subsequently, to an impaired functioning of the entire organism (Figure 1) [9]. A diet deficient in iodine can lead to mental retardation, hypothyroidism, congenital anomalies, goitre or low IQ, whereas the iodine excess can result in iodine-induced hyperthyroidism [6]. Therefore, an adequate iodine intake—150 μg/day for adults, 120 μg/day for children, and 250 μg/day for pregnant women—is essential for the proper thyroid gland function [3]. The aim of the study is to evaluate and review diets in search of the best diet to protect against iodine deficiency. Moreover, in this paper we wanted to assess which populations are at risk of iodine deficiency or excess. In order to collect the literature related to the presented topic, the PubMed database (www.pubmed.ncbi.nlm.nih.gov, accessed on 20 January 2021) was explored with reference to the terms “iodine”, “diet”, “deficiency”. Iodine Global Network materials and reports were also analyzed. We have focused on popular diets—for example; paleolithic, gluten free or vegan diets—and investigated if individuals following them may be at risk of iodine deficiency. The risk of iodine deficiency may be increased due to the type of products that are eliminated, for example dairy. However, we also discussed if there are any food substitutes, which could decrease this risk. Further, salt-restrictive and plant-based diets may be associated with iodine deficiency due to reduction or elimination of salt and animal derived foods, respectively. We also mentioned low-salicylate diet, gluten-free diet and parenteral nutrition that are associated with many nutritional deficiencies.

## 2. Iodine Deficiency

### 2.1. Iodine Deficiency among Various Populations

According to the WHO, iodine deficiency is defined by a median urinary iodine concentration < 100 µg/day [11]. On the basis of the Iodine Global Network report from 2019, which included school-age children, 115 countries were classified as countries with the optimal level of iodine supply (the United States of America, Canada, France, Portugal, Great Britain) and in 23 countries the iodine intake was too low—Burkina Faso, Burundi, Cambodia, Finland, Germany, Israel, Iraq, Lebanon, Madagascar, Morocco, Haiti, North Korea, Mozambique, Nicaragua, Samoa, Tajikistan, Vanuatu, Norway, Russia, South Sudan, Sudan, and Vietnam. In Angola and Italy, iodine intake was considered to be sufficient (Figure 2) [12]. As it has already been mentioned, around 2 billion are at risk of iodine deficiency, according to the WHO [13]. However, it should be noted that the report was based on the UIC results of school-age children, and thus, it cannot reflect the status of iodine, for instance, among pregnant women. Moreover, the data regarding the status of iodine in different populations remain insufficient. [14]. Most of the analyzed articles demonstrate an emerging iodine deficiency in the population of women of childbearing age, which indicates a public health issue concerning the improvement of iodine status of the abovementioned group of women [15,16]. Gizak et al. emphasised that an insufficient intake of iodine among pregnant women persists, since iodine intake among pregnant women is insufficient in 39 out of 72 countries [17].

Pregnant and breastfeeding women are particularly exposed to iodine deficiency, and, therefore, iodination of salt could not be sufficient, and additional supplementation should be considered [18].

### 2.2. Health Consequences of Iodine Deficiency

Health consequences of iodine deficiency can affect all ages—adults, adolescents, children, infants and foetuses. Similarly, goitre formation and susceptibility to the nuclear radiation affect all age groups. In terms of adults, iodine deficiency has been associated with an impaired mental function, reduced work output, goitre and hypothyroidism. In children and adolescents, on the other hand, it has been linked to mental and growth retardation. Moreover, an increase in the incidence of papillary thyroid cancer (PTC) compared to follicular thyroid cancer (FTC) has been observed, amounting to 0.19 to 1.7, respectively [19]. It is now believed that the introduction of the iodine supplementation in subjects with a significant iodine deficiency may lead to an increase in the PTC/FTC ratio [20,21]. The transition from FTC to PTC may be associated with an elevated rate of the V600E BRAF (B-Raf serine/threonine kinases) mutation over time [22].

### 2.3. Iodine Deficiency and Breast Cancer

The cause–effect relationship between iodine deficiency, impaired function of the thyroid gland, and breast cancer has been observed for over 100 years [23,24,25]. Iodine is absorbed via sodium/iodide symporter in the glandular breast tissue, and its role is to promote the growth of the normal tissue [26]. In fact, research studies have demonstrated the role of iodine as an antioxidant agent in the mammary gland [27,28]. Thus, it has been suggested that iodine deficiency can increase the risk of breast cancer in two mechanisms. The first is associated with direct influence on the glandular breast tissue, which could be explained by means of an increased sensitivity to oestradiol in the case of iodine deficiency. The other is associated with an impaired thyroid gland function and, therefore, hypothyroidism as the consequence of iodine deficiency [26]. Numerous studies demonstrate the association between hypothyroidism, especially autoimmune, and breast cancer; however, this relationship remains controversial [24,29,30,31,32,33,34]. Significantly, an excessive intake of iodine, by stimulating activity of ER-α, negatively affects the risk of breast cancer [35]. Simultaneously, an impaired thyroid gland function can result in a slower tumour growth and in challenging the diagnosis with an increased tumour invasiveness [26]. In Japan, almost 3-times lower incidence of breast cancer was observed, when compared with the USA; nevertheless, the incidence of breast cancer among Japanese women living in the USA was similar to other women. It suggests that the iodine-rich diet could be preventive in the development of breast cancer. Therefore, diets comprising a low amount of iodine could be associated with an increased risk of breast cancer development, which emphasises the need to decrease iodine deficiency in all age groups.

### 2.4. Iodine Deficiency in Pregnant Women

The adequate level regarding the requirements of iodine is particularly important in the population of women in the childbearing age. It should be noted that not only severe maternal iodine deficiency, but also mild to moderate deficiencies, have been associated with the consequences appearing in the offspring.

Levie et al. conducted a meta-analysis with an individual participant data from three prospective population-based cohorts in order to investigate the relationship between the maternal iodine status during pregnancy and the infant IQ in 6180 mother-child pairs from three countries. The aim of the study was to identify the sensitive time windows of exposure to the suboptimal iodine availability. The study demonstrated that mild to moderate iodine deficiencies were associated with a reduction in verbal IQ scores of children, especially in the first trimester of pregnancy [36].

Additionally, the results obtained in the Avon Longitudinal Study of Parents and Children indicate that there is an inverse correlation between the low maternal iodine status in the first trimester and the offspring IQ scores at the age of eight, as well as reading accuracy, comprehension, and reading score at the age of nine years. The study comprised 1040 pregnant women and their children in the UK [37].

In the study by Gietka-Czernel et al. [38], which involved 100 pregnant women between the 5th and 38th week of pregnancy with normal thyroid function, only 35% of the subjects took iodine supplements, and 59% were on a diet rich in iodine carriers. Moreover, the presence of goitre was found in 28% of the pregnant women. In fact, iodine deficiency can also lead to a spontaneous abortion, stillbirth, birth cretinism, congenital disabilities, maldevelopment of the foetal brain and an increased risk of perinatal mortality [39,40].

## 3. Iodine Recommended Intake

### 3.1. Iodine Intake from the Diet, Iodine Fortification and Recommendations

Iodine intake correlates with its blood concentration, therefore, the proper iodine intake is essential [41]. However, it seems that iodine fortification is necessary in order to meet the daily requirements. It has been observed that a higher consumption of iodine-fortified bread was associated with a more frequent consumption of proper iodine amounts [42].

Although salt iodisation is recommended, iodisation of salt is not obligatory in several countries, e.g., the United States of America [43]. Salt iodination is mandatory in Poland, China or Denmark but is voluntary in Holland or Great Britain. Interestingly, in Australia and New Zealand, salt iodination is not mandatory, nevertheless iodised salt is used in the baking of bread. The level of salt iodination varies among countries from 8 to 100 mg of iodine/kg of salt [44]. Daily recommended allowances (RDA) and adequate intake (AI) are presented in Table 1.

### 3.2. Food Sources of Iodine

Fish, seafood, milk, dairy, vegetables, and fruits are considered as a good source of iodine. However, fish are usually not consumed frequently enough to cover the daily iodine requirements [47]. The amount of iodine in vegetables and fruits depends on the type of soil in which they had been planted [45] and content of iodine in soil is different in various world regions. Iodine content in marine plants is higher than in terrestrial plants [48] Moreover, the content of iodine in milk and dairy also varies from 200 μg/L even to 1000 μg/L [49].

There is a number of factors affecting iodine concentration in milk, e.g., farm management and animal keeping (outdoor or indoor) which are associated with the iodine intake [50]. Moreover, the concentration of iodine in organic milk is lower, as compared to the conventional milk. It is vital to notice that iodine concentration in milk does not depend on the fat content [51]. Moreover, the content of iodine in milk was significantly higher in winter than in the summer, due to the fact that milk yield is highest in the summer-autumn months (it tends to be calving time). Therefore, the concentration of microelements is lower [52]. Mullan et al. also reported that concentration of iodine in milk is higher in the winter than in the summer. Additionally, urinary iodine concentration among girls was lower in summer than winter and was positively correlated with milk consumption [53]. In fact, there are differences between iodine concentrations in milk between various regions in the winter and in the summer which suggest that iodine concentration is dependent on feeding [54]. According to Hejtmánková et al., the content of iodine in the environment affects iodine concentration in milk [55].

Although it is recommended to limit the salt intake, it still remains the primary source of iodine, e.g., in Slovenia iodised salt was the primary source of iodine for adolescents. However, it should be noted that even though iodine intake was appropriate, the salt consumption was exceeded, thus, a decrease of the salt intake must have been associated with other nutritional interventions in order to cover daily iodine requirements [56]. Iodine content in the selected food products is presented in Table 2 to summarize and compile data on iodine content in selected products.

## 4. Diets and a Reduced Iodine Intake

### 4.1. Hypertension and Salt-Restrictive Diet

At the end of the 20th century, an increased prevalence of cardiovascular disease (CVD) was observed mostly in the developed countries. It was associated with such factors as lifestyle changes, a decreased physical activity and an increased consumption of processed food with a high content of salt, fat, and energy. As an increased mortality due to CVD was observed, prevention programmes were initiated, including a recommendation to reduce salt intake, since it was associated with an increased risk of hypertension, stroke, atherosclerosis, or several types of cancers [10,60]. In 2006 and 2007, during WHO expert consultations, a limited intake of salt was recommended which amounted to maximum 5 g of salt per day (2 g of sodium). In contrast, in Poland about 13.5 g of salt is consumed daily, which could seem high when compared to other European countries. However, in other non-European countries, it can even reach 20 g of salt per day [7,10,61]. It seems somewhat ironic that a limited salt consumption is recommended, when it was previously used as a tool aimed at the prevention of iodine deficiency. Nevertheless, the WHO indicates that iodination of salt is economically beneficial, and iodine does not influence the salt taste [62]. Currently, it is recommended to fortify 1 kg of salt with 20–40 mg of iodine, depending on the actual salt consumption in a particular country [7]. Moreover, taking Poland as an example, 6.5 g of salt should be consumed to provide the recommended iodine intake for adults (150 µg/day). Therefore, it could be easily observed that the consumption of other sources of iodine is essential in order to meet the iodine and salt requirements. Additionally, it should be vital among individuals who limit their salt consumption, or those who follow any type of elimination diet [61].

### 4.2. Vegan and Vegetarian Diets

Currently, vegan and vegetarian diets are gaining greater recognition. A vegetarian diet excludes meat, fish and seafood (the exclusion of dairy and eggs varies depending on the vegetarian diet type). On the other hand, a vegan diet, as more radical type, excludes all animal products. Properly balanced vegan and vegetarian diets are considered safe to follow at every stage of life, even during pregnancy and infancy [63]. However, incorrectly composed vegetarian diets can lead to a deficiency of protein, unsaturated omega-3 fatty acids, and several vitamins and minerals, such as vitamin B12, vitamin D, calcium, zinc, iron, or iodine [64]. As it has been previously mentioned, milk and dairy products constitute the sources of iodine the elimination of which could lead to iodine deficiency among vegans and vegetarians. In fact, marine algae are the primary source of iodine in vegetarian diets [65]. Vegan and vegetarians who do not include marine algae in diet, or do not supplement iodine are more susceptible to iodine deficiency than individuals following a less restrictive diet [63]. Moreover, a high consumption of soy has been observed among individuals following vegetarian diets, which is the source of protein, iron, zinc or group B vitamins [66]. In contrast, according to the in vitro studies, soy isoflavones, i.e., genistein and daidzein, can affect the thyroid peroxidase function [65,67]. In addition, soy can also negatively affect the thyroid gland function when iodine is deficient [65,66]. Nevertheless, it should be noted that thermal processing eliminates most of the soy goitrogens, and their consumption should not be discouraged in order to prevent iodine deficiency. Therefore, it would seem that limited consumption of cruciferous vegetables and soy could be considered among the hypothyroid individuals with a low iodine intake [64,65].

### 4.3. Gluten-Free Diet

A gluten-free diet needs to be followed by individuals with coeliac disease. It is associated with the elimination of wheat, rye, barley, and oat (in certain cases). On the other hand, it is recommended to consume naturally occurring gluten-free grains, vegetables (including pulses), fruits, meats and fishes, dairy, and the substitutes for traditional gluten products (with gluten content not exceeding 20 mg/kg) [68,69]. Vici et al. noticed that a gluten-free diet was usually low in protein, vitamin D and B12, folic acid, iron, zinc, magnesium, and calcium [70]. Additionally, it was observed that individuals following a gluten-free diet consume inadequate amounts of selenium, riboflavin, niacin, and thiamine, with a simultaneous high consumption of fat, carbohydrates, and sodium [71]. However, according to the current studies, a gluten-free diet is not associated with an increased risk of iodine deficiency [69].

### 4.4. Iodine Intake and Dairy Foods

As it has been mentioned, dairy and milk products, among others, are sources of iodine. Although the consumption of milk and dairy products varies between countries and populations, they remain one of the most important dietary sources of iodine, and can contribute to about 13–64% of the daily requirement [72,73]. In the UK, Norway, and France milk and dairy consumption account for 38%, 60%, and 40% of the iodine intake, respectively [74,75,76]. Additionally, dairy products should be consumed often, particularly in those countries where iodine-fortification of salt is not mandatory, or where the availability of iodised salt is limited. However, it is worth bearing in mind that the amount of iodine in dairy and milk products differs and depends on seasonality, farming practice, milk processing, and the fortification of animal feeds [77].

Dairy intake is usually below recommended levels, although the current dietary guidelines recommend the consumption of 2–4 serving per day of fat-free or low-fat dairy products. However, the recommendations are country-specific and hence may vary [78]. Low dairy consumption can be associated with an inadequate intake of several minerals, such as calcium, magnesium, potassium, iodine or vitamins, such as vitamin D [79]. Although milk and dairy products were rich in iodine in the Israeli study, the population’s intake was insufficient due to a low dairy and milk consumption [77]. Furthermore, a lower consumption of dairy products was associated with a lower iodine intake when compared with consuming two or more dairy portions per day among pregnant women [80]. In the Little in Norway Study (LiN), dairy consumption was associated with a urinary iodine concentration, or urinary iodine to creatinine ratio among pregnant women; however, the total iodine intake was still not in accordance with the current recommendations [75]. On the other hand, although pregnant women in Australia were iodine sufficient, less than 50% of them were able to meet the estimated average requirement (EAR) from food alone without supplementation, with dairy as the primary source of dietary iodine [81]. Iodophors (iodine-containing sanitisers) are used in the dairy industry and are associated with the iodine content in milk and dairy. Although iodine-contaminated milk has been a significant source of iodine in several countries, e.g., in Australia, the use of iodophors declined and resulted in lower iodine concentrations in milk, which resulted in a lower overall iodine intake [82]. In fact, milk and its products are more frequently replaced by non-cow’s milk and plant-based products, which are scarcely fortified in iodine, possibly leading to iodine deficiencies among vegetarian individuals [83]. It is crucial to notice that adding seaweed to the plant-based products, in order to fortify them in calcium, can increase the iodine contents, although the iodine concentration depends on the type of seaweed extract used.

### 4.5. Parenteral Nutrition

Although the oral absorption of iodine is high—around 90%—and efficient, parenteral nutrition (PN) can be associated with the risk of iodine deficiency due to the low iodine content in the parenteral formulas [84]. However, a routine supplementation of PN formulas with iodine generally is not recommended and varies between the guidelines. On the one hand, thyroid gland stores enough iodine to meet the needs for less than three months and the long-term patients with PN can consume enough iodine by means of a standard diet (assuming the proper absorption of iodine in the duodenum) [85,86]. On the other hand, iodine is administered in PN formulas in Europe, although not in the USA; however, the American Society for Parenteral and Enteral Nutrition (ASPEN) recommended changing the commercially available parenteral multivitamin and multi-trace element products, including iodine supplementation [87]. In the study by Willard et al., the investigated PN solutions contained a small amount of iodine (iodine amount was not included on the solutions’ labels), and the patients receiving the long-term PN would require about 5.6 L/day of a formula to meet the recommended daily allowance (RDA), whereas the standard administration of PN formulas is around 1.8–2.4 L/day [88]. However, as Navarro et al. study demonstrated, patients with the short bowel syndrome were able to meet the recommended intake of iodine with the standard diet and did not present changes in the iodine status [89]. Furthermore, previously used povidone-iodine solutions for skin cleansing, resulting in the transcutaneous iodine absorption, were replaced by chlorhexidine-based antiseptics [87]. Findik et al. indicated that the use of povidone iodine solutions during the caesarean sections influenced free triiodothyronine (fT3) thyroid-stimulating hormone (TSH) values in the mother, although it has not statistically significantly influenced urine iodine levels [90]. Additionally, it was also reported that use of iodine-containing antiseptics increased the urinary iodine status, without influencing the TSH status in the neonate [91]. The use of iodine-containing solutions as a dermal disinfectant in newborns is strongly discouraged and their exposure to povidone iodine during caesarean procedures should be monitored for possible thyroid disorders [92]. Moreover, in terms of infants, iodinated disinfectants and adventitious iodine content in PN solutions were sufficient and additional supplementation was unnecessary. However, due to the current use of not iodinated antiseptics, iodine supplementation should be considered [86,93].

### 4.6. Palaeolithic Diet

A palaeolithic-type diet (PtD), typically used for the bodyweight reduction, is based on products consumed before the agriculture was introduced; thus, in such a diet foods like processed oils, dairy, grains, legumes, salt or refined sugars are eliminated. On the other hand, the consumption of lean meats, nuts, eggs, fruits, and vegetable is high [94]. Even though meat can constitute a source of iodine, the study of Franke et al. study demonstrated that iodine content in pork was low, even if the feed was iodine-enhanced [95]. Taking that into consideration the abovementioned fact, individuals following the PtD can be at risk of iodine deficiency. According to Manousou et al., PtD has been associated with the risk of mild iodine deficiency and, as the authors have suggested, iodine supplementation should be considered [96]. Similar results were found in the study by Genoni al., where a PtD group showed a lower intake of several micronutrients, including iodine, than a group following healthy eating recommendations. Nevertheless, as the authors noted, the use of salt, or the issue of whether it was iodised, could be underestimated [94]. As the Churuangsuk et al. systematic review revealed, the intake of iodine decreased after following any carbohydrate-restricted diet (CRD). Since carbohydrate-rich products do not constitute the main source of iodine, it could be suggested that this observation was associated with a restricted intake of dairy products in CRD [97]. However, in view of iodine intake, not only can the number of carbohydrates matter, but also their quality. As Louie et al. study showed, Australian children and adolescents, who consumed carbohydrates from high-glycemic-index products, presented a significantly higher risk of not meeting the iodine requirements [98]. If our ancestors’ diet was low in carbohydrates and lower in iodine than the current diets, the question arises concerning how it could influence the current iodine requirements, and whether they are linked to iodine deficiency disorders (IDD). Interestingly, changes in the human nutrition due to the industrial and agricultural development resulted in increased triiodothyronine (T3) levels, and furthermore, an increased T3 production caused an elevated iodine requirements [99]. Thus, following PtD can currently be misleading and associated with an increased risk of IDD.

### 4.7. Low-Salicylate Diet

A low-salicylate diet eliminates salicylates from food products, due to the unfavourable reaction of the human body to acetylsalicylic acid. Several fruits and vegetables, marinated or dried products, as well as honey, most of the spices, apple vinegar, nuts, fruit and vegetable juices, and fruit syrups are rich in salicylates [100,101,102]. Therefore, considering the number of food products, which must be eliminated, a low-salicylate diet could increase the risk of nutritional deficiencies. According to Szczuko et al., iodine intake was insufficient and lower than the recommended values among men and women who followed a low-salicylate diet [102]. In fact, the greater the difference between the recommended and actual dose, the smaller the number of calories—52.19 μg, 37.78 μg, and 12.27 for the diet of 1500 kcal, 2000 kcal, and 2500 kcal, respectively. However, the differences could be explained by the fact that the higher amounts of calories were consumed, the higher the overall food intake, including iodine-rich products. Additionally, the development of the IDD due to the restricted diets was observed in other cases [103,104,105]. Hence, it is essential to assess the nutritional intake, including iodine intake, among individuals following the restricted diets, due to the potential risk of nutritional deficiencies.

## 5. Excessive Intake of Iodine

### 5.1. Excessive Intake of Iodine among Various Populations

The excessive consumption of iodine is much less frequently observed. However, it is mainly witnessed in the Asian countries, such as Japan or Korea, where for over 13 thousand years, marine algae have been essential in the local cuisine [106,107]. The European Food Safety Authority (EFSA) and the WHO consider the values adopted by the Scientific Committee on Food (SCF) and the Institute of Medicine (IOM) regarding the tolerable upper limit for iodine uptake, i.e., 600 µg/d and 1100 µg/d respectively [13,46,108,109]. However, in Japan, the average intake of iodine is around 500–1000 µg/day, and in some region it reaches almost 20,000 µg/d [107,110]. Although most of the population can tolerate high doses of iodine, due to the internal self-regulation mechanisms in the thyroid gland, cases of iodine-associated impaired thyroid gland function were observed [111]. The risk of excessive iodine intake is mostly associated with the consumption and supplementation of marine algae in which iodine content is highly varied, depending on the type and the geographical origin; it could fluctuate between 0.06 mg/100 g to 624.5 mg/100 g of dry mass [107,112]. Additionally, the lack of iodine content on the product label is also problematic [112]. For instance, the average consumption of marine algae in Japan is around 4–7 g per day [107]. Moreover, it should be noted that due to the increased salt consumption, a recommended daily intake of iodine can also be exceeded and, therefore, the role of salt iodination and iodine status in the population should be highlighted [106]. It is also recommended to carefully apply supplements based on marine algae, particularly in pregnant women where the use of such supplements should be discouraged [112,113]. Iodine excess can also lead to thyrotoxicosis, e.g., the Jod-Basedow phenomenon which occurs in the course of treatment or remission of Graves’ disease or the tuberous goitre in people living in the areas of iodine deficiency [114]. Furthermore, the consumption of proper amounts of iodine is significant for a developing foetus. In fact, an increased demand for this mineral in pregnant women is caused by an increased production of thyroid hormones by half, the passage of iodine through the blood-placenta barrier, and an increased renal clearance of this element [18,115]. In the case of pregnancy and breastfeeding, the need for iodine in the form of oral supplements should be increased to a dose of 220–290 μg per day. However, the American Thyroid Association recommends a daily intake of 150 µg of iodine to ensure a normal development of the brain and the nervous system of the foetus and the baby [116]. According to the WHO, an excessive iodine intake (UIC > 500 μg/L) may also have a detrimental effect on the thyroid function, leading to subclinical hypothyroidism, while UIC of 150–249 μg/L is an adequate iodine intake for pregnant women, although not for an individual patient [117,118]. Moreover, the incidence of thyroid cancer has increased, which is associated with several and diverse factors, for example, increased iodine supplementation or obesity [115,119]

### 5.2. Pharmacotherapy and Excessive Iodine Intake

Pharmacological doses of iodine—even 1000 times higher than a daily requirement—are found in many medical substances, e.g., iodine-containing contrast media (140–400 mg of iodine per 1 mL), disinfectants (10 mg of iodine per 1 mL), amiodarone (one 200 mg tablet contains 75 mg of iodine), or Lugol’s solution (127 mg of iodine in 20 drops) [120]. The healthy thyroid gland has a defence mechanism against the excessive iodine intake, called the Wolff–Chaikoff effect, which temporarily inhibits the biosynthesis of thyroid hormones [110,121]. The structural formula of amiodarone, a benzofuran derivative, is similar to the T4. A molecule of this substance contains two atoms of iodine, which represents around 37.5% of its mass, 10% of which is deionised daily to the iodine. Therefore, of 200–400 mg of amiodarone daily, 6–12 mg of iodine is supplied, hence, if 150–200 µg of iodine is considered a the daily requirement, the iodine demand is exceeded multiple times [122]. The lack of abovementioned defence mechanisms can lead to hypothyroidism among individuals treated with amiodarone [121].

## 6. Summary

Both the deficiency as well as the excessive intake of iodine can lead to adverse health consequences. The group which is most at risk of iodine deficiency, due to the increased requirements, are pregnant women. It should be noted that the increasing prevalence of breast cancer could also be associated with a lower iodine intake, thus, monitoring of iodine intake should be one of the breast cancer prevention measures. Several diets, such as a vegan, vegetarian, palaeolithic, or low salicylate diet, are also associated with an increased risk of iodine deficiency, however in fact, any improperly balanced diet can result in iodine deficiency. Interestingly, several societies recommend limiting the consumption of dairy fat, therefore, possibly affecting dairy consumption overall, which could also have an impact on the iodine intake. The global recommendation to iodise salt resulted in significant improvements in the iodine status globally. For instance, introducing iodised salt resulted in the eliminating of IDD in school-age children in China, or the incidence of goitres in several areas in India [123,124]. However, certain issues remain unresolved. Limited salt consumption is recommended as a prevention of various non-communicable diseases, such as hypertension, stroke, or atherosclerosis. Moreover, a consumption of processed foods, which do not necessarily include iodised salt, also affects the iodine intake and its status. It is worth bearing in mind that achieving an adequate iodine status among pregnant women can result in an excessive iodine status among children. Therefore, it seems necessary to find other alternatives, apart from the iodised salt, in order to prevent iodine deficiency. However, in several countries, particularly those with a lower income, universal salt iodization programs should be introduced and promoted in order to prevent goitres and IDD. Furthermore, it is vital to remember that no perfect diet exists, fully covering the iodine requirements, which suggests that iodine fortification and supplementation is crucial.

## Figures and Tables

**Figure 1 nutrients-13-00513-f001:**
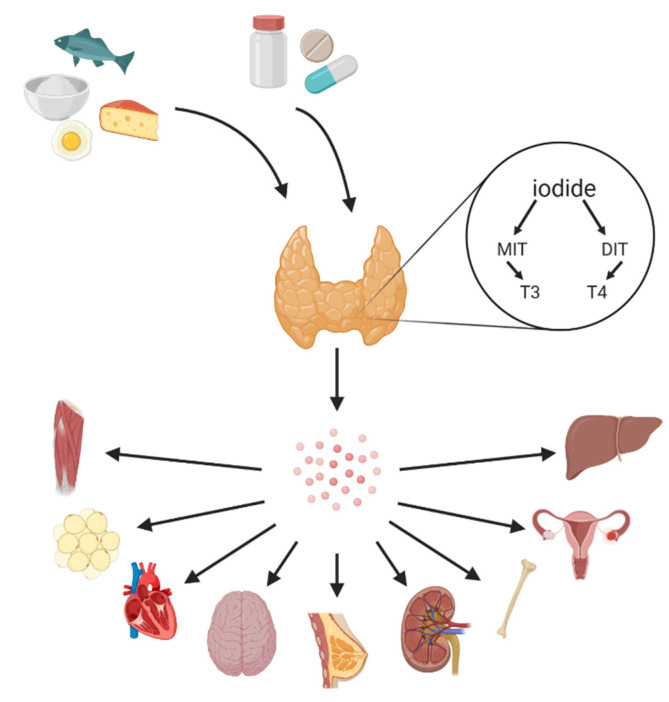
Iodine can be ingested through the diet and dietary supplements. The primary sources of iodine are salt, seafood, fish, algae, milk, and dairy. Iodine is taken up by the thyroid gland and used for synthesis MIT (monoiodotyrosine) and DIT (diiodotyrosine), which are used for thyroid hormone (triiodothyronine and thyroxine) biosynthesis. Receptors of thyroid hormones are on the surface of different organs; thus, thyroid hormones travel in the blood and may affect many systems among other reproductive system, skeletal or muscular system.

**Figure 2 nutrients-13-00513-f002:**
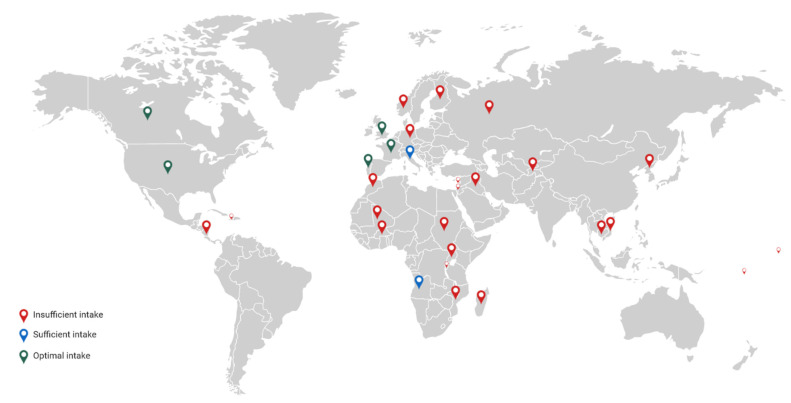
Insufficient iodine intake is observed in many regions, mainly African and Asian countries. Optimal intake of iodine occurs among other in the United States, Canada, and some countries in Europe. However, intake of iodine is not adequate in many European regions.

**Table 1 nutrients-13-00513-t001:** Recommended dietary allowances and adequate Intake for iodine [45,46].

Group	RDA (Recommended Dietary Allowances), (μg)	Adequate Intake (AI), (μg)
0–6 months		110
7–12 months		130
1–3 years	90	
4–8 years	90	
9–13 years	120	
14–18 years	150	
≥19 years	150	
Pregnancy	220	
Breastfeeding women	290	

**Table 2 nutrients-13-00513-t002:** Iodine content in the selected products. [10,57,58,59].

Product	Iodine Content (μg/100 Grams of Product)
Codfish, fresh	110
Salmon, fresh	7.7–44
Pike, fresh	8
Kefir, 2% of fat	7.5
Skimmed milk	19.5–21
Cheese, full-fat	7.7–30
Chicken eggs	9.5–57.6
Oat flakes	0.5–5.9
Vegetable	1–31
Nuts	4–9
Salt, iodized	2293

## Data Availability

Statement excluded.

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
