# Peer review of "Is There an Ideal Diet to Protect against Iodine Deficiency?"

_nutrients, 2021, doi:10.3390/nu13020513_

Round 1
Reviewer 1 Report
This manuscript sets out to investigate whether any diet is suitable to protect against iodine deficiency. While various diets are discussed, I feel that the manuscript adopts a broad-brush approach to the topic and, as such, lacks sufficient detail.
In the abstract, the authors suggest that global salt iodisation began in 1980. In fact, iodised salt was first introduced by the pioneering American doctor David Marine in the1920s to mitigate iodine deficiency in the so-called goitre-belt of the USA. Subsequently, iodised salt was discredited and withdrawn but was re-introduced in several countries in the 1950s and 1960s. The use of iodised salt resulted in the effective eradication of endemic goitre from the developed world by the 1970s, to the extent that it was described in a Scientific American article in the early 1970s as “a disease of the poor”.
The re-emergence of iodine deficiency in the developed world probably reflected changes in diet. It is possible that this may reflect, in part, limitation of salt in the diet, but is equally likely to reflect lower consumption of dairy produce, due to suggestions from the medical profession that consumption of dairy produce was a major cause of heart disease etc.
I would suggest that the authors need to expand the discussion on dietary sources of iodine. As is pointed out in the manuscript iodised salt is a major source of dietary iodine, there needs to be greater discussion of this. Use of iodised salt in China has almost eradicated iodine deficiency in that country, mainly due to the cooking of vegetables in water containing iodised salt. Similarly, in India the major source of dietary iodine is vegetables cooked in water containing iodised salt. In other countries where iodised salt is available problems occur due to the consumption of food prepared without iodised salt, e.g., bread.
The authors point out that there is considerable variation of iodine content of dairy produce etc. I feel that this needs to be discussed in greater detail. There is a considerable body of data that details iodine contents of various foods in different countries, I feel that this is not reflected in the single table included in the manuscript. As indicated in the manuscript, iodine in dairy produce reflects supplementation of feed together with the use of I-containing sterilants in milking parlours. In many countries cattle are kept indoors during the winter and consume I-supplemented feed, as a result milk produced during winter is richer in iodine. A recent study in Ireland (Mullan et al., 2019), found that urinary iodine content of girls (14-15) was lower in summer than in winter, which correlated with milk consumption.
Studies have also shown that in countries where there is no I-supplementation of feed, there is a geographical variation in I content of milk, with that deriving from cows grazing in coastal areas producing milk with far higher contents of I than those grazing in upland areas more remote from the sea. In addition, it has been shown that leafy vegetables grown in near-coastal locations are richer in iodine than those grown further iland.
There are also some other points I would make. In the discussion of iodine deficiency among populations, it is suggested that the data available cannot assess the problems of iodine deficiency in pregnant woman. Gisak et al. (2018) report on the global iodine status of pregnant women.
In lines 329-330, it is suggested that salt in the USA is not iodised, this is not true, iodised salt is available.
Reviewer 2 Report
Dear Authors,
Your manuscript entitled “Is there an ideal diet to protect against iodine deficiency?” offers a very good review concerning the role of iodine as essential nutrient to maintain a normal thyroid function as well as a healthy metabolism.
The article is well written, providing an interesting insight into the significance of diet as the main contributor to iodine status, focusing on the more consumed diets worldwide.
However, I have found some points that require to be clarified:
- Line 99: the cause-effect relationship has been studied, analysed or explored but not conducted. Please, replace the verb to achieve a more understandable statement.
- Line 103: some studies have demonstrated the role of iodine as antioxidant agent in the mammary gland, such as: a)Venturi S. Is there a role for iodine in breast diseases? Breast 2001 Oct;10(5):379-82. DOI: 10.1054/brst.2000.0267 PMID: 14965610. b) Smyth PP. Role of iodine in antioxidant defence in thyroid and breast disease. Biofactors. 2003;19(3-4):121-30. doi: 10.1002/biof.5520190304. PMID: 14757962
- Lines 233-235: iodinated antiseptic solutions contain high concentrations of iodine likely to block the immature thyroid gland of neonates, so currently their use in caesarean sections and NICU units is strongly discouraged. a) Findik RB, Yilmaz G, Celik HT, Yilmaz FM, Hamurcu U, Karakaya J. Effect of povidone iodine on thyroid functions and urine iodine levels in caesarean operations. J Matern Fetal Neonatal Med. 2014 Jul;27(10):1020-2. doi: 10.3109/14767058.2013.847417. PMID: 24060143
- Lines 278-279: The reference of the EFSA panel resolution should be added. EFSA Journal 2014;12(5):3660.
- Lines 295-301 and lines 304-309 are not related to iodine excess but iodine metabolism in physiological conditions. They may be better located in point 2.4 (pregnant women).
- Lines 309-310: The incidence of thyroid cancer is raising all over the world, due to several and diverse factors: a) Kim J, Gosnell JE, Roman SA. Geographic influences in the global rise of thyroid cancer. Nat Rev Endocrinol. 2020 Jan;16(1):17-29. doi: 10.1038/s41574-019-0263-x. PMID: 31616074
- Reference 85 is outdated, and must be replaced by the current ATA guidelines: a) Alexander EK, Pearce EN, Brent GA, Brown RS, Chen H, Dosiou C, Grobman WA, Laurberg P, Lazarus JH, Mandel SJ, Peeters RP, Sullivan S. 2017 Guidelines of the American Thyroid Association for the Diagnosis and Management of Thyroid Disease During Pregnancy and the Postpartum. Thyroid. 2017 Mar;27(3):315-389. doi: 10.1089/thy.2016.0457. PMID: 28056690
- Line 355: I guess that “civilisational” diseases meant noncommunicable or cardiovascular diseases.
Reviewer 3 Report
- This is a useful review/opinion piece that draws attention to population groups at risk of iodine deficiency and highlights challenges associated with specific dietary practices.
- The review is extensive and comprehensive drawing on over 100 references.
- Whist useful by drawing attention to high risk groups and the challenges associated with correcting iodine deficiency using diet, the paper does not offer anything new by way of analysis, or provide any novel solutions.
- As a review, there is no methods section describing how the studies/literature referenced were identified.
- Minor errors include:
- In a number of places commas rather than full stops have been used to signify a decimal point - see lines 268 and 285.
- Line 280 average iodine intake is around 500-1000 5µg/day makes no sense - suspect there is a typographical error.
Round 2
Reviewer 1 Report
The manuscript has been improved considerably with most of my comments dealt with adequately. However, I am still concerned with the discussion of dietary iodine. Table 2 presents data for iodine in chosen foodstuffs, however, it is derived from a single publication and is composed of single figures, i.e. not ranges and not for different countries. In particular, I would point out that the concentration of iodine in iodised salt shows considerable variation. There are several publications which present data on dietary iodine sources, these could provide data which could supplement the table. I would also suggest that the authors consult the paper by Fuge and Johnson (2015) in Applied Geochemistry.
Reviewer 2 Report
The authors have addressed each one of my suggestions. The manuscript has substantially changed and improved.
No further comments.
Reviewer 3 Report
Thank you for the opportunity to comment on this revised manuscript. In its current form, this remains an opinion piece backed by a comprehensive body of literature.
- The purpose of the paper is not clear. In the introduction is states the aim is to assess which populations are at risk of deficiency or excess iodine while the title suggests the purpose is to identify whether there is an ideal diet to protect against deficiency. The purpose could be clearer and better related to the summary.
- The only method referred to is a literature search on the topic. It would strengthen the manuscript if the rationale for the choice of specific restrictive diets was justified and the methods used to undertake the analysis of these diets was described. Are these the most popular restrictive diets followed????
- How was the analysis of the adequacy of such diets undertaken and how do they compare to non-restrictive diets??? Vegan diets might actually be high in iodine if seaweed is regularly consumed and likewise with gluten free and palaoe diets if seafood is regularly consumed.
- It would be more logical if the section on food sources of iodine preceded the analysis of specific restrictive diets.
- Ref for Table 1 is a secondary source. This should be a primary source.
- Ref for Table 2 (ref 10) should this be (ref 9)???
- The purpose for Table 2 is not clear in the text.
- As a suggestion, a rewrite focussing on a desktop analysis of popular restrictive diets and the impact on dietary iodine intake might make this more publishable. Some of the background could be condensed and more information on why certain diets have been chose and how they were assessed would be valuable. A comparison with non-restrictive typical diets would also be valuable.
